# The Learning Curve of Robotic Thyroid Surgery and the Avoidance of Temporary Hypoparathyroidism after Total Thyroidectomy and Concomitant Central Compartment Node Dissection: A Single Surgeon's Experience

**Jae Hyun Park [1], Jun Hyeok Lee [2], Jae Won Cho [3] and Jong Ho Yoon [1,\*]**

[1]   Department of Surgery, Wonju Severance Christian Hospital, Yonsei University Wonju College of Medicine, Wonju 26426, Korea

[2]   Department of Biostatistics, Wonju Severance Christian Hospital, Yonsei University Wonju College of Medicine, Wonju 26426, Korea

[3]   Department of Surgery, Asan Medical Center, University of Ulsan College of Medicine, Seoul 138-736, Korea

\*   Correspondence: gsyoon@yonsei.ac.kr

**Abstract:** The aim of this study was to evaluate the learning curve of robotic thyroid surgery with regard to both operation time and temporary hypoparathyroidism using quantitative statistical analysis. A total of 194 patients who underwent total thyroidectomy and concomitant central compartment node dissection for papillary thyroid carcinoma by a single surgeon between December 2008 and September 2017 were enrolled. The learning curve for operation time was assessed using the cumulative sum (CUSUM) technique, and the number of procedures required to reduce the incidence of temporary hypoparathyroidism to less than 30% was determined using the CUSUM and risk-adjusted CUSUM (RA-CUSUM) techniques. The learning curve for operation time was divided into three phases: phase 1 (the initial learning period, 1st–19th cases), phase 2 (the challenging period, 20th–121st cases), and phase 3 (the competent phase, 122nd–194th cases). To reduce the incidence of temporary hypoparathyroidism to <30% required 119 cases, and after adjustment for potential risk factors by RA-CUSUM analysis this extended to 173 cases. Technical proficiency for robotic thyroid surgery with respect to the avoidance of surgical complications probably requires a longer learning period than that required for operation time.

**Keywords:** learning curve; robotic thyroid surgery; papillary thyroid carcinoma; total thyroidectomy; transient hypoparathyroidism

## 1. Introduction

New surgical technologies have associated learning curves, and operation times and surgical complication rates are considered measures of the learning process [1–4]. The learning curve represents an approximation of how surgical outcomes change with experience. Based on an estimated learning curve, we predicted the number of patients required for a surgeon to achieve proficiency in a particular operative technique which was determined by standard cut-off points. The learning curve in surgery can be defined as the number of cases required to perform the procedure with reasonable operating time and an acceptable rate of complications, resulting in an adequate postoperative clinical outcome [2].

In the field of thyroid surgery, the learning curve of robotic surgery for differentiated thyroid carcinoma has been previously studied [5–10]. However, these studies had several limitations. First, learning curves of robotic thyroid surgery were produced for only operation time, and variables, such

as, blood loss, surgical failure, perioperative complication, and others were not included. Moreover, only the moving average method was used and clinicopathological risk factors that might affect the learning curve were not considered.

In this study, we aimed to evaluate the learning curves of robotic thyroid surgery for operation time and surgical complications using quantitative statistical analysis in a cohort of papillary thyroid carcinoma (PTC) patients who underwent robot-assisted total thyroidectomy (TT) and concomitant central compartment node dissection (CCND).

## 2. Materials and Methods

### 2.1. Patient Population

The medical records of 196 consecutive patients who underwent robot-assisted TT and concomitant CCND using a gasless trans-axillary approach for PTC as an initial operation between December 2008 and September 2017 were retrospectively reviewed. After excluding two patients with unavailable data, 194 patients were enrolled. The study protocol was approved beforehand by Asan Medical Center institutional review board, and the requirement for informed patient consent was waived because of the non-interventional nature of the study.

Baseline clinicopathologic features, including parameters related to technical outcomes and surgical completeness, were reviewed. Technical outcomes were evaluated with respect to operation time (defined as time from skin incision to completion of skin closure) and surgery-related complication rates. Surgical completeness was assessed using numbers of retrieved lymph nodes and serum stimulated thyroglobulin (sTg) levels at time of radioactive iodine (RAI) remnant ablation (ablation sTg) and 6~12 months after remnant ablation (control sTg).

### 2.2. Robotic Procedures and Instruments

Robotic gasless trans-axillary TTs with CCNDs were performed by the double-incision approach. In brief, the patient was placed in the supine position under general anesthesia with the neck slightly extended. A 1 side arm was placed on an arm board and extended the cephalad to expose the axilla. A 5 to 6 cm vertical skin incision was made along the lateral border of the pectoralis major muscle in the axilla. Dissection was performed by electrocautery above the pectoralis major muscle to create a space using increasingly long retractors to elevate the skin, subcutaneous tissue, and platysma. After exposing the medial border of the sternocleidomastoid (SCM) muscle, the dissection was performed using the avascular space between the sternal and clavicular heads of the SCM muscle and beneath the strap muscle until the contralateral lobe of the thyroid was exposed. A 0.8 cm second skin incision was made for fourth robotic arm insertion on the anterior chest wall on the tumor side (2–4 cm superiorly and 6–8 cm medially to the nipple, and away from the sternum). Robotic docking was performed after placing the external retractor. A 30 degree down dual-channel telescope, controlled by the surgeon, was then placed at the center of the incision. A Maryland dissector and harmonic-curved shears (both from Intuitive Surgical) were then placed on both lateral ends of the axillary incision. A pair of Prograsp forceps (Intuitive Surgical) was then inserted through a second incision on the anterior chest wall and placed laterally to the sternal branch of the SCM muscle. The entire time from the patient's positioning to the robotic docking was about 30 min. The thyroid gland was retracted using a Prograsp forceps on the fourth robotic arm and the dissection was performed using harmonic curved shears and a Maryland dissector. This procedure allowed the surgeon to use 3 robotic arms during robot-assisted thyroidectomy. All procedures were carried out by a single endocrine surgeon (JH Yoon) using the da Vinci Si or Xi intuitive surgical system (Sunnyvale, CA, USA). The surgical team for these robotic procedures consisted of only three members, such as an operator, an assistant nurse, and a scrub nurse.

The surgeon (JH Yoon) has been performing more than 100 conventional thyroidectomies annually in an academic setting since 2003. The surgeon had performed more than 350 endoscopic thyroidectomies and was fully trained in several basic endoscopic procedures. Prior to attempting the

robotic thyroidectomy, an instruction video of the experienced robotic surgeon was provided to the surgeon and a regular surgical skill training program was created by the experienced robotic surgeon. The surgeon had watched more than 5 cases of robotic thyroidectomy of the experienced robotic surgeon without participating. The experienced robotic surgeon helped the surgeon while performing more than 2 cases of robotic thyroidectomy, prior to the surgeon performing independently.

### 2.3. Surgical Complications

Surgery-related complications investigated in the present study included postoperative hemorrhage, wound infection, temporary and permanent hypoparathyroidism, and recurrent laryngeal nerve (RLN) injury. Postoperative hemorrhage was defined as the requirement for hematoma evacuation. Hypoparathyroidism was defined as: a postoperative serum calcium level of < 8.0 mg/dL (reference, 8.6–10.2 mg/dL) and/or the presence of hypocalcemic symptoms and signs. Patients with an ongoing need for calcium or vitamin D supplementation to maintain normocalcemia or to relieve hypocalcemic symptoms and signs beyond 6 months after initial operation were classified as having permanent hypoparathyroidism, and others were classified as having temporary hypoparathyroidism. The perioperative complication with the highest rate was chosen as the target complication for the learning curve analysis.

### 2.4. Learning Curve of Robotic Thyroid Surgery

The learning curve for robotic total thyroidectomy was assessed with regard to operation time and the incidence of target complication. Before and after the measure assessed by the learning curve analysis, baseline clinicopathologic characteristics and surgical completeness variables were compared.

### 2.5. Inclusion and Exclusion Criteria of Robotic Thyroid Surgery for PTC

Before 2014, the inclusion criteria of robotic thyroid surgery using a gasless trans-axillary approach with double incision were as follows: (1) PTC of ≤2 cm in maximum diameter and (2) minimal invasion to the anterior thyroid capsule and strap muscle. However, in 2014, the inclusion criteria were expanded to include a primary tumor size up to 4 cm. Exclusion criteria were not altered during the study period and were as follows: definite posterior capsular invasion particularly adjacent to the tracheoesophageal groove, lateral lymph node metastasis, and distant metastasis at presentation. All patients were informed whether they were candidates for robotic surgery based on considerations of inclusion and exclusion criteria, and were then informed of costs, given brief details of the surgical procedure, possible complications, and expected outcomes of robotic and conventional open surgery. Patients were then allowed to choose a surgical option.

### 2.6. Surgical Strategy in Patients with Thyroid Cancer

Before 2016, TT was recommended for almost all patients with a primary tumor of >1 cm, based on the 2009 American Thyroid Association (ATA) guidelines [11]. However, since 2016, the surgical paradigm shifted to more conservative surgery based on the 2015 ATA guidelines [12], and therefore, TT was mainly recommended for patients with a primary tumor of >4 cm, bilateral cancers, gross extrathyroidal extension (T4 classification), and lateral lymph node metastasis found during preoperative evaluation or operation.

At our institution, CCND is routinely performed for patients with thyroid cancer; at a minimum, unilateral prophylactic CCND is performed even in patients without suspicious lymph nodes on preoperative imaging studies or during surgery. Even though the primary cancers are bilateral, the prophylactic CCND is performed unilaterally on the dominant lesion side only, unless there are suspicious lymph nodes or enlarged lymph nodes. Bilateral CCND is only performed therapeutically. Compartment-based nodal dissection, not "berry picking", is performed in preference.



*2.7. Postoperative Follow-up Protocol*

All patients received follow-up examinations at an outpatient clinic. Patients who underwent TT without RAI remnant ablation were placed on thyroid-stimulating hormone (TSH) suppression therapy and followed up at 1, 6, and 12 months, and annually thereafter. Thyroid function tests were routinely performed at every visit, and neck ultrasonography was performed annually. In patients who underwent TT with RAI remnant ablation, a diagnostic whole-body scan (WBS) following thyroid hormone withdrawal or recombinant human TSH (rh-TSH) administration was performed 6–12 months after remnant ablation with the simultaneous measurement of control sTg. Serum Tg/anti-Tg antibody measurement and neck ultrasonography were performed on all patients during follow-up. When control sTg was ≥1 ng/mL and neck ultrasonography showed no evidence of disease, $^{18}$F-deoxyglucose positron emission tomography or chest computed tomography were considered to localize persistent or remnant disease. Any patient suspected of having loco-regional recurrence underwent ultrasonography-guided fine needle aspiration cytology (FNAC) or core needle biopsy (CNB).

Structural recurrence was defined as the appearance of cytologically or histopathologically proven malignant tissue, or the appearance of highly suspicious structural lesions on cross-sectional or functional imaging studies after a minimum period of 1 year of no evidence of disease (NED) after initial treatment. Persistent structural disease was defined as the appearance of structural lesions without a period of NED for one year after the initial treatment.

*2.8. Statistics*

The statistical analysis was performed using SAS Ver. 9.4 (SAS, Cary, NC, USA) and R Ver. 3.4.3 (The R Foundation for Statistical Computing, Vienna, Austria). Categorical variables are presented as numbers and percentages, and continuous variables as means ± standard deviations or medians with ranges. Categorical variables were analyzed using the Chi square test or Fisher's exact test, and continuous variables using the independent two-sample *t*-test. *P*-values < 0.05 were considered statistically significant. To analyze the learning curve for operation time, the cumulative sum (CUSUM) technique was used. The learning curve for target complication was analyzed using the CUSUM and risk-adjusted CUSUM (RA-CUSUM) techniques. In the RA-CUSUM analysis, an acceptable rate of target complication was determined to be the mean of levels previously reported. The potential risk factors associated with the development of target complication were selected by multivariate logistic regression with backward elimination. Parameters with *p*-values < 0.15 were considered potential risk factors and kept in the final model. Multivariate logistic regression was used to calculate the probability of target complication with adjustment for potential risk factors.

2.8.1. Cumulative Sum (CUSUM)

The CUSUM is a statistical technique which shows the sequential difference between the individual data and the mean value [13]. In this study, the CUSUM technique was applied by the following equation, where $x_i$ is an individual datum and $\mu$ is the mean value.

$$\text{CUSUM} = \sum_{i=1}^{n}(x_i - \mu)$$

In terms of surgical performance, CUSUM allows investigators to visualize normally indiscernible data trends and judge whether variations in performance are acceptable [14]. In the present study, CUSUM was used as a method to plot the learning curve with respect to both operation time and target surgical complication.

2.8.2. Risk-Adjusted CUSUM (RA-CUSUM)

The RA-CUSUM can be regarded as an extension of CUSUM and reflects risk factors throughout the likelihood-based score [15,16]. The RA-CUSUM demonstrates cumulative differences between

expected and actual occurrence of a surgical complication. For each patient, the probability of a surgical complication was determined using a multivariate logistic regression model, which in turn determined the rate at which the plot of ascends or descends [9]. The RA-CUSUM values were calculated using the following equation, in which $x_i$ is defined as the presence of surgical complication, $\tau$ is the reported acceptable surgical complication rate, and $P_i$ is the probability of surgical complication in each case.

$$\text{RA} - \text{CUSUM} = \sum_{i=1}^{n} (x_i - \tau) + (-1)^{x_i} P_i$$

When a target complication occurs, $x_i$ is scored as 1, but if no target complication occurs, $x_i$ is scored as 0. The RA-CUSUM moves upwards when a target complication occurs or downwards when no target complication occurs. In this study, the RA-CUSUM method was used to assess the minimum number of cases required to ensure technical proficiency in terms of the incidence of target complication after adjusting for potential risk factors.

## 3. Results

### 3.1. Baseline Clinicopathologic Features

Baseline clinicopathologic features of 194 patients who received robotic TT with CCND during the study period (December 2008 to September 2017) are summarized in Table 1. Mean patient age was 41.8 ± 8.8 years, and the cohort was composed of 187 females and 7 males. The mean tumor size was 1.0 ± 0.6 cm. Multiple and bilateral thyroid gland lesions were observed in 90 (46.4%) and 70 (36.1%) patients, respectively. In all, 163 (84%) patients underwent ipsilateral CCND and 31 patients had bilateral CCND. Mean postoperative hospital stay was 3.8 ± 1.7 days.

There were no significant differences between the before ($n$ = 119) and after ($n$ = 75) the learning curve end-point with regard to primary tumor size ($p$ = 0.084), extrathyroidal extension ($p$ = 0.518), lymph node metastasis ($p$ = 0.219), and postoperative hospital stay ($p$ = 0.305).

When we compared the multiplicity, bilaterality, and extent of LN dissection between the before ($n$ = 119) and after ($n$ = 75) learning curve end-point, the multiplicity (35.3% versus 64%, $p$ < 0.001), bilaterality (26.1% versus 52%, $p$ < 0.001), and bilateral CCND (10.1% versus 25.4%, $p$ = 0.005) were more frequent after ($n$ = 75) the learning curve end-point, respectively.

All operations were successfully completed using a robotic technique without conversion to conventional open surgery. The mean follow-up period was 76.1 months (range 6–103 months).

**Table 1.** Baseline clinicopathologic features of patients who underwent robotic total thyroidectomy with concomitant central compartment node dissection, overall, before, and after the learning curve end-point.

| Characteristics | Overall ($n$ = 194) | Before ($n$ = 119) | After ($n$ = 75) | $p$-Value |
|---|---|---|---|---|
| Age, year, mean ± SD | 41.8 ± 8.8 | 41.9 ± 8.4 | 41.6 ± 9.4 | 0.788 |
| Sex, $n$ (%) | | | | 0.045 |
| Male | 7 (3.6) | 7 (5.9) | 0 (0) | |
| Female | 187 (96.4) | 112 (94.1) | 75 (100.0) | |
| Primary tumor size, cm, mean ± SD | 1.0 ± 0.6 | 0.9 ± 0.5 | 1.1 ± 0.6 | 0.084 |
| Extrathyroidal extension, $n$ (%) | | | | 0.518 |
| None | 73 (37.6) | 42 (35.2) | 31 (41.3) | |
| Minimal | 77 (39.7) | 51 (42.9) | 26 (34.7) | |
| Extensive | 44 (22.7) | 26 (21.9) | 18 (24.0) | |
| Multiplicity, $n$ (%) | 90 (46.4) | 42 (35.3) | 48 (64.0) | <0.001 |
| Bilaterality, $n$ (%) | 70 (36.1) | 31 (26.1) | 39 (52.0) | <0.001 |
| Extent of LN dissection, $n$ (%) | | | | 0.005 |
| Ipsilateral CCND | 163 (84.0) | 107 (89.9) | 56 (74.7) | |
| Bilateral CCND | 31 (16.0) | 12 (10.1) | 19 (25.4) | |
| Lymph node metastasis, $n$ (%) | 85 (43.8) | 48 (40.3) | 37 (49.3) | 0.219 |
| Postoperative hospital stay, day, mean ± SD | 3.8 ± 1.7 | 3.9 ± 1.8 | 3.7 ± 1.6 | 0.305 |

SD = standard deviation; LN = lymph node; CCND = central compartment node dissection.

### 3.2. Technical Outcomes of Robotic Thyroid Surgery

Technical outcomes with regard to operation time, blood loss, and the rate of perioperative complications are described in Table 2. Mean operation time for robotic TT with CCND was 174.9 min and the mean blood loss was 4.3 ± 3.2 mL. The most common complication was temporary hypoparathyroidism with an incidence of 36.6%, and thus, this was chosen as the target complication for the learning curve analysis. Permanent hypoparathyroidism and intraoperative recurrent laryngeal nerve injury occurred in 1.0% and 1.5% of patients, respectively. Inadvertent injury to the RLN was treated by Rofilan injection laryngoplasty immediately after surgery. Surgical site infection and chyle leakage occurred in 1.0% and 2.6% of patients, respectively, and all responded to conservative management within a few days. There was no postoperative hemorrhage that required re-operation.

**Table 2.** Technical outcomes of robotic thyroid surgery.

| Characteristics | Number of Patients (*n* = 194) |
|---|---|
| Operation time, min, mean ± SD | 174.9 ± 36.1 |
| Blood loss, ml, mean ± SD | 4.3 ± 3.2 |
| Postoperative hemorrhage, *n* (%) | 0 (0) |
| Temporary hypoparathyroidism, *n* (%) | 71 (36.6) |
| Permanent hypoparathyroidism, *n* (%) | 2 (1.0) |
| RLN injury, *n* (%) | 3 (1.5) |
| Infection, *n* (%) | 2 (1.0) |
| Seroma, *n* (%) | 0 (0) |
| Chyle leakage, *n* (%) | 5 (2.6) |
| Tracheal injury, *n* (%) | 0 (0) |

SD = standard deviation; RLN = recurrent laryngeal nerve.

### 3.3. Surgical Completeness of Robotic Rhyroid Surgery

The surgical completeness of robotic TT with CCND was evaluated in terms of the number of retrieved lymph nodes and ablation and control sTg levels (Table 3). The mean number of retrieved LNs retrieved was seven. Of the 194 study subjects, 159 (82.0%) underwent RAI remnant ablation. After excluding patients with positive anti-Tg Ab levels, 133 and 108 patients were evaluated for ablation and control sTg levels, respectively. The proportions of patients with an ablation sTg < 10 ng/mL or a control sTg < 1 ng/mL were 94.0% and 91.7%, respectively. The surgical completeness presented as the number of retrieved LNs and ablation/control sTg levels did not significantly differ between before and after the learning curve end-point (120th case).

**Table 3.** Surgical completeness of robotic thyroid surgery, overall, before, and after the learning curve end-point.

| Characteristics | Overall (*n* = 194) | Before (*n* = 119) | After (*n* = 75) | *p*-Value |
|---|---|---|---|---|
| Retrieved LNs, *n*, median (range) | 7 (0–24) | 6 (0–24) | 8 (0–24) | 0.135 |
| RAI remnant ablation, *n* (%) | 159 (82.0) | 107 (89.9) | 52 (69.3) | <0.001 |
| Anti-Tg Ab (+), *n* (%) | 30 (15.5) | 21 (17.7) | 9 (12.0) | 0.289 |
| Ablation sTg, ng/mL *, median (range) | 0.55 (0.08~47.10) | 0.54 (0.08–47.10) | 0.76 (0.08–17.80) | 0.899 |
| Ablation sTg < 10 g/mL *, *n* (%) | 125/133 (94.0) | 81/88 (92.1) | 44/45 (97.8) | 0.259 |
| Control sTg, ng/mL, median (range) | 0.08 (0.08~16.50) | 0.08 (0.08–16.50) | 0.08 (0.08–6.30) | 0.872 |
| Control sTg < 1 ng/mL, *n* (%) | 99/108 (91.7) | 68/74 (91.9) | 31/34 (91.2) | 1.000 |

SD = standard deviation; LN = lymph node; RAI = radioactive iodine; Anti-Tg Ab = anti-thyroglobulin antibody; sTg = stimulated thyroglobulin. * Calculated only for patients with negative anti-Tg Ab levels.

*3.4. Learning Curve for Operation Time and the Incidence of Temporary Hypoparathyroidism*

The CUSUM analysis of operation time showed two peaks at the 19th and 121st cases (Figure 1), and CUSUM analysis of the incidence of temporary hypoparathyroidism showed a peak at the 119th case, from which the incidence of temporary hypoparathyroidism following robotic TT began to decline to <30% (Figure 2), which was determined by calculating the mean value of previously reported values [17–27]. In addition, multivariate logistic regression with backward elimination identified age, gender, primary tumor size, multifocality, lymph node metastasis, extent of LN dissection, and thyroid weight as potential risk factors of temporary hypoparathyroidism ($p < 0.15$) (Table 4). Male gender was additionally corrected using FIRTH's bias correction method. The RA-CUSUM analysis after adjusting for these potential risk factors showed that the incidence of temporary hypoparathyroidism fell below 30% at the 173rd case (Figure 2).

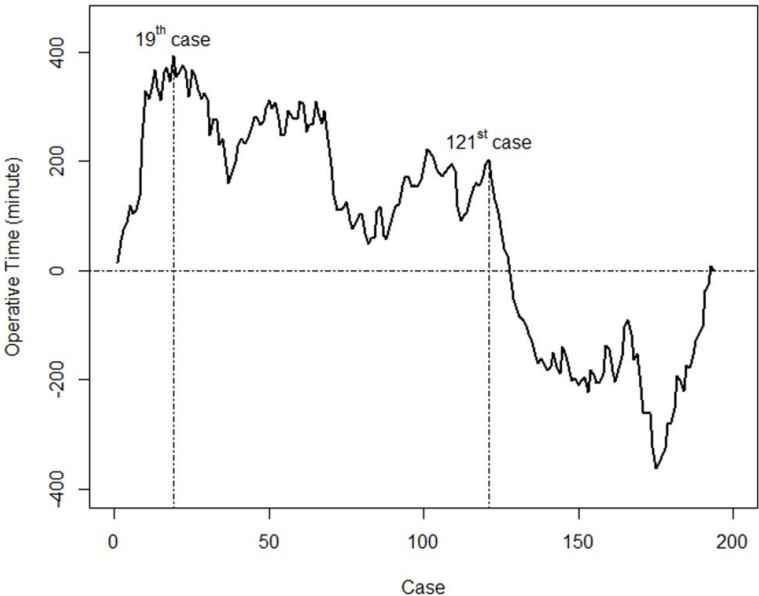

**Figure 1.** The learning curve for operation time as determined by the cumulative sum (CUSUM) analysis.

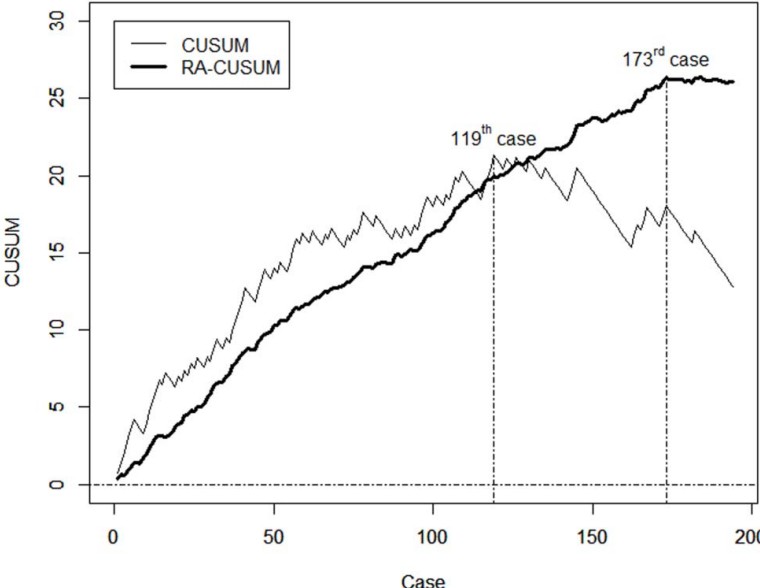

**Figure 2.** The learning curves for temporary hypoparathyroidism as determined by CUSUM and risk-adjusted CUSUM (RA-CUSUM) analysis.

**Table 4.** Potential risk factors associated with the development of temporary hypoparathyroidism as determined by backward selection ($p < 0.15$).

| Characteristics | Number of Patients ($n = 194$) | |
| --- | --- | --- |
| | Odds Ratio (95% CI) | *p*-Value |
| Age, years | 0.98 (0.94–1.02) | 0.285 |
| Gender * | | |
| Female | Ref | |
| Male | 3.83 (0.70–20.87) | 0.121 |
| Primary tumor size, cm | 0.55 (0.29–1.07) | 0.078 |
| Multifocality | | |
| No | Ref | |
| Yes | 0.57 (0.30–1.08) | 0.083 |
| Lymph node metastasis | | |
| No | Ref | |
| Yes | 1.86 (0.93–3.73) | 0.078 |
| Thyroid weight, gram | 1.04 (0.99–1.09) | 0.136 |

Ref = reference. * This parameter was corrected using FIRTH's bias correction because of the small proportion of males.

*3.5. Comparison of Basic Clinicopathologic Features and Surgical Completeness before and after the 120th Case*

Clinicopathologic features of patients who underwent robotic surgery after the 120th case revealed lower proportion of males, more multifocal and bilateral cases, and a higher proportion of bilateral CCND than in patients before the learning curve end-point (Table 1). Surgical completeness as assessed using the number of retrieved LNs and ablation and control sTg levels was not significantly different before versus after the 120th case (Table 3).

## 4. Discussion

In the present study, the learning curve of robotic TT with concomitant CCND with regard to operation time was divided into three phases: phase 1 (the initial learning period, 1st–19th case), phase 2 (the challenging period, 20th–121st case), and phase 3 (the competent phase, 122nd–194th case). The learning curve with regard to perioperative complication showed that 119 cases were required to maintain a temporary hypoparathyroidism rate < 30%. However, after adjusting for potential risk factors, this case number increased to 173.

Several studies have evaluated the learning curves of different surgical procedures [13,28–31]. In these studies, learning curves were analyzed with respect to rates of surgical failure or perioperative complications as well as the operation time using quantitative statistical methods [13,28–31]. The learning curve for robotic thyroid surgery with regard to the operation time has been evaluated several times since robotic surgery was introduced to the field of thyroid surgery in South Korea in 2007 [5–10]. The first assessment of the learning curve for robotic thyroidectomy using a gasless trans-axillary approach by a single surgeon showed that the operation times for robotic thyroidectomy reached a baseline after approximately 40 to 45 operations [5]. Regarding robotic thyroidectomy using bilateral axillo-breast approach (BABA), the surgeons completed the learning curve after performing 40 surgeries [10]. Also, in a multicenter study, the operation times were markedly decreased after completion of 50 robotic total thyroidectomies with central compartment node dissection [6]. However, these earlier studies had several limitations in that the learning curve was evaluated by mixing total thyroidectomy and lesser than total thyroidectomy [5,10], with respect to the operation time [5,6,10]. Moreover, previous studies [5,6,10] (70–100 patients) did not enroll the sufficient number of patients to accurately determine the long-term learning curve in robotic total thyroidectomy. In the present study, the learning curve of robotic thyroid surgery was evaluated in terms of operation time and the incidence of temporary hypoparathyroidism, which is the most common and challenging surgical complication following TT, by applying the CUSUM and RA-CUSUM methods to the data of

patients treated by TT and concomitant CCND. In addition, potential risk factors for development of temporary hypoparathyroidism were selected and adjusted. To the best of our knowledge, the present study is the first to evaluate the learning curve of robotic thyroid surgery with respect to a perioperative complication.

In the present study, the learning curve for robotic thyroid surgery with regard to the operation time consisted of three phases. The initial learning phase (1st–19th cases), during which additional time is required for creating a working space and robotic docking. Unlike intra-abdominal surgery, the neck area has no preformed space, and thus, additional time is required for working space creation and robotic docking. However, this additional time can be easily shortened by the accumulation of experience. From the 20th to the 121st cases (the challenging phase) operation times fluctuated presumably because, during this period, robotic procedures were applied to more complex and challenging cases than those during the initial learning period. During the third phase (the competent phase; 122nd–194th cases), surgical proficiency was accomplished and stably maintained. However, the rising trend of operation time around the 170th case in the third phase should be further evaluated with more abundant data, even though it was thought to be due to the application of the robotic surgery to more complex cases with an accumulation of surgical experience. Furthermore, it was noticeable that the CUSUM curve for the incidence of temporary hypoparathyroidism peaked at the 119th case, which was close to the second peak on the operation time CUSUM curve at the 121st case. The surgical proficiency gained by advanced anatomical understanding of feeding vessels to and venous drainage from parathyroid glands, meticulous dissection, individual ligation, appropriate handling of energy devices, and ensuring a safe distance to avoid thermal injury to parathyroid glands, obtained during the challenging phase, probably contributed to the lower incidence of temporary hypoparathyroidism observed during the competent phase. Despite a greater proportion of multifocal, bilateral, and bilateral CCND cases in the competent phase than in the challenging phase, the operation time and the incidence of temporary hypoparathyroidism during the competent phase rather declined, reflecting the achievement of surgical proficiency. The RA-CUSUM analysis showed that the learning period was greater for patients with risk factors for temporary hypoparathyroidism. On the other hand, surgical completeness as determined using the number of retrieved LNs and ablation and control sTg levels was stably maintained during the entire study period.

The present study had several limitations that warrant consideration. First, it was subject to inherent bias because of its retrospective nature. Second, the present study did not consider any variables associated with the surgical team besides the operator. The learning curve for robotic thyroid surgery might also rely on the proficiency of the surgical team as well as the operator. Therefore, those variables should be considered to more accurately assess the learning curve for robotic thyroid surgery, and the formal training program for them should be prepared. Third, our institutional preference for prophylactic CCND, even in non-invasive PTC, may also be debatable because it is not recommended by international guidelines [11,12,32]. This preference is based on our opinion that prophylactic CCND is not a time-consuming procedure, does not increase surgical complications, and allows more accurate staging and initial risk stratification. In the present study, the incidence of perioperative surgical complications was maintained at a low level, and thus, we believe the receipt of prophylactic CCND did not affect results. Forth, the follow-up period was too short to directly evaluate the long-term oncologic safety of robotic thyroid surgery with regard to structural recurrence. Fifth, we did not consider body mass index as a parameter for risk adjustment, because the prevalence of obesity is much lower in South Korea than in the West. Accordingly, caution should be exercised when applying our results to populations with high obesity rates. Furthermore, we suggest further study is required to determine the effect of body mass index on perioperative complications.

In conclusion, technical proficiency at robotic TT with CCND in PTC patients with respect to avoiding temporary hypoparathyroidism is probably achieved after a learning period of around 120 cases. However, this learning period is probably longer when patients have potential risk factors associated with development of temporary hypoparathyroidism.

**Author Contributions:** J.H.P. and J.H.Y. conceived and designed the study; J.W.C. acquired the data. J.H.L. analyzed the data; J.H.P. and J.H.Y. wrote the paper. All the authors contributed to this paper.

**Funding:** This research received no external funding.

**Conflicts of Interest:** The authors declare no conflicts of interest.

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
