# Peer review of "The Learning Curve of Robotic Thyroid Surgery and the Avoidance of Temporary Hypoparathyroidism after Total Thyroidectomy and Concomitant Central Compartment Node Dissection: A Single Surgeon’s Experience"

_applsci, doi:10.3390/app9132594_

Round 1

Reviewer 1 Report

The paper is written well. It is a summary of the learning curve of a robotic thyroid surgery from a single surgeon’s experience. The main contribution is to introduce surgical complications in the statistic data.

The overall structure of the paper is clear and precise. The presentation of the content is organised well and easy to follow. Following concerns should be addressed in the revision:

1.      The introduction is too short. It should include more analysis of the previous work. Such as previous surgeon’s experience and whether previous work has reported the complications in other ways. Is there any difference of the their robotic thyroid surgery to yours?

2.      Give more background on the robotic surgery. Which robot did you use? Which instruments did you used? What are other supporting devices?  

3.      Give more background of surgeon. Did he get training before the thyroid surgery? How many conventional thyroid surgeries did he do before? Did he operate other surgery? How many years of these experience?

4.      In the material and method part, the operation team should have more information. Such as how many assistive surgeons? What is the set-up time? How much did the supportive team have robotic training before?

5.      In 2.7.2. suggest using other variables rather than x_i in RA_CUSUM equation because it has different meaning from the equation in CUSUM. For the value of P_i, it is case-dependant variable. How did you define its value?

6.      In the section 3.1, 3.2 and 3.3, the discussion is too short. Please give more discussion, at least three paragraphs for each table.

7.      For the learning curve for operation time, it will be very interesting to see the comparison to other literature and to conventional thyroid surgery. A single source data is always weak. The absolute operation should also be shown. In figure 1, the rising of the operation time after around 170 cases should be explained.

8.      The y-axis label of the figure 2 should be temporary hypoparathyroidism case.

9.      Did you ever have report of abandoned robotics cases?

10.   In figure 2, the RA-susum value is decreasing after 119th case. Is it due to the surgeon get familiar to the robotic system and the risk value P_i is estimated lower?

Author Response

Answers to reviewers’ comments

Thank you for giving us the opportunity to revise our manuscript. My colleagues and I feel that the quality of our article has greatly improved because of the reviewers’ insightful comments, and we have provided ‘point-by-point’ replies to each comment below.

Reviewer 1

1.  The introduction is too short. It should include more analysis of the previous work. Such as previous surgeon’s experience and whether previous work has reported the complications in other ways. Is there any difference of their robotic thyroid surgery to yours?

R: Thank you for your insightful comments. A learning curve represents an approximation of how surgical outcomes change with experience. Based on an estimated learning curve, we predicted the number of patients required for a surgeon to achieve proficiency in a particular operative technique which was determined by standard cut-off points. The learning curve in surgery can be defined as the number of cases required to perform the procedure with reasonable operating time and an acceptable rate of complications, resulting in an adequate postoperative clinical outcome. [2] The learning curve for robotic thyroid surgery regarding the operation time that has been evaluated several times since robotic surgery was introduced to the field of thyroid surgery in South Korea in 2007. [5-10] The first assessment of learning curve for robotic thyroidectomy via gasless transaxillary approach by a single surgeon showed that the operation times reached a baseline after approximately 40 to 45 operations. [5] Regarding robotic thyroidectomy using bilateral axillobreast approach (BABA), the surgeons overcame the learning curve after performing 40 surgeries. [10] Also, in multicenter studies, the operation times were markedly decreased after completion of 50 robotic total thyroidectomies with central compartment node dissection (CCND). [6] However, these earlier studies had several limitations that the learning curve was evaluated by mixing total thyroidectomy and lesser than total thyroidectomy, [5, 10] with respect to the operation time. [5, 6, 10] Moreover, previous studies [5, 6, 10] (70-100 patients) did not enroll the sufficient number of patients to more accurately determine the long-term learning curve in robotic total thyroidectomy. We have added these contents in the ‘Introduction’ and ‘Discussion’ sections.

2.      Give more background on the robotic surgery. Which robot did you use? Which instruments did you used? What are other supporting devices?  

R: Thank you for your constructive comments. Robotic gasless transaxillary TT with CCND was performed by the double-incision approach. In brief, the patient was placed in the supine position under general anesthesia with the neck slightly extended. A 1-side arm was placed on an arm board and extended the cephalad to expose the axilla. A 5- to 6-cm vertical skin incision was made along the lateral border of the pectoralis major muscle in the axilla. Dissection was performed by electrocautery above the pectoralis major muscle to create a space using increasingly long retractors to elevate the skin, subcutaneous tissue and platysma. After exposing the medial border of the sternocleidomastoid (SCM) muscle, the dissection was performed using the avascular space between the sternal and clavicular heads of the SCM muscle and beneath the strap muscle until the contralateral lobe of the thyroid was exposed. A 0.8 cm second skin incision is made for fourth robotic arm insertion on the anterior chest wall on the tumor side (2–4 cm superiorly and 6–8 cm medially to the nipple, and away from the sternum). Robotic docking is performed after placing the external retractor. A 30-degree-down dual channel telescope, controlled by the surgeon, is then placed at the center of the incision. A Maryland dissector and harmonic curved shears (both from Intuitive Surgical) are then placed on both lateral ends of the axillary incision. A Prograsp forceps (Intuitive Surgical) is then inserted through a second incision on the anterior chest wall and placed laterally to the sternal branch of the SCM muscle. The thyroid gland is retracted using a Prograsp forceps on the fourth robotic arm and the dissection is performed using a harmonic curved shears and a Maryland dissector. This procedure allows the surgeon to use 3 robotic arms during robot-assisted thyroidectomy. All procedures were carried out by a single endocrine surgeon (JH Yoon) using the da Vinci Si ot Xi intuitive surgical system (Sunnyvale, CA, USA). We have added these in the ‘Materials and Methods’ section as the ‘Robotic procedure and instruments’.

3.      Give more background of surgeon. Did he get training before the thyroid surgery? How many conventional thyroid surgeries did he do before? Did he operate other surgery? How many years of these experience?

R: Thank you for your comments. In a recent study of patients undergoing thyroidectomy in the Health Care Utilization Project Nationwide Inpatient Sample (HCUP-NIS), surgeons were divided into low- (<10 cases/yr; encompassing 6072 surgeons), intermediate- (10-100 cases/yr; 11,544 surgeons), and high-volume (>100 cases/yr; 4009 surgeons) categories. Surgeries included in the present study were all performed by a single endocrine surgeon (JH Yoon). The surgeon (JH Yoon) has been performing more than 100 conventional thyroidectomies yearly in an academic setting since 2003. The surgeon had performed more than 350 endoscopic thyroidectomies and was fully trained in several basic endoscopic procedures. Prior to attempting the robotic thyroidectomy, an instruction video of the experienced robotic surgeon was provided to the surgeon and a regular surgical skill training program was created by the experienced robotic surgeon. The surgeon has watched more than 5 cases of robotic thyroidectomy of the experienced robotic surgeon without participating. The experienced robotic surgeon helped the surgeon while performing more than 2 cases of robotic thyroidectomy, prior to the surgeon being independent. We have also added these in the ‘Materials and Methods’ section as the ‘Robotic procedure and instruments’.

4.      In the material and method part, the operation team should have more information. Such as how many assistive surgeons? What is the set-up time? How much did the supportive team have robotic training before?

R: Thank you for your insightful comments. The surgical team for these robotic procedures consisted of only three members, such as an operator, an assistant nurse, and a scrub nurse. The entire time from the patient’s positioning to the robotic docking was about 30 minutes. We have also added these in the ‘Materials and Methods’ section as the ‘Robotic procedure and instruments’. As you mentioned, the learning curve for robotic thyroid surgery might also rely on the proficiency of the surgical team as well as the operator. Therefore, variables associated with the proficiency of surgical team should be considered to more accurately assess the learning curve for robotic thyroid surgery, and the formal training program for them will be needed. However, we unfortunately don’t have any formal training program for them yet. We have added these as one of the limitation of this present study in the ‘Discussion’ section.

5.      In 2.7.2. suggest using other variables rather than x_i in RA_CUSUM equation because it has different meaning from the equation in CUSUM. For the value of P_i, it is case-dependant variable. How did you define its value?

R: Thank you for your insightful comments. In CUSUM, there is a difference between the continuous variable and binary variable.

In a continuous variable, CUSUM represents the cumulative sum of observations – average. In binary variable, CUSUM represents the cumulative sum of observed value (0 or 1) – observation probability. RA-CUSUM is adjusted by adding a risk-adjusted probability in the binary case of CUSUM.

In RA-CUSUM, x_i is 1 when an event occurs and 0 if it does not. P_i is risk-adjusted probability. In order to control variables which affecting outcome, we select variables that affect outcome through logistic regression with backward elimination. The probability value (when event=1) obtained from this logistic regression model is P_i.

6.      In the section 3.1, 3.2 and 3.3, the discussion is too short. Please give more discussion, at least three paragraphs for each table.

R: Thank you for your comments. We have revised the section 3.1, 3.2 and 3.3.

7.  For the learning curve for operation time, it will be very interesting to see the comparison to other literature and to conventional thyroid surgery. A single source data is always weak. The absolute operation should also be shown. In figure 1, the rising of the operation time after around 170 cases should be explained.

R: Thank you for your insightful comments. We have added the results from other literatures in ‘Discussion’ section. Unfortunately, we had not compared the learning curve for robotic thyroid surgery with conventional open surgery. Please consider our limitations. Additionally, we thought the rising trend of operation time after 170th case was probably because at this time point robotic surgery was applied to more complex and challenging cases with an accumulation of surgical experience. It looks like further studies will be needed with more abundant data. We have added this in the ‘Discussion’ section.

8.      The y-axis label of the figure 2 should be temporary hypoparathyroidism case.

R: Thank you for your comments. The y-axis is cumulative sum of observed value (0 or 1) – observation probability for temporary hypoparathyroidism. For example, if the prevalence of an event is 30% and an event occurs in the first case, CUSUM = 1 – 0.3 = 0.7. In the second case, no event occurs, CUSUM = 1 – 0.3 + 0 – 0.3 = 0.4.

9.      Did you ever have report of abandoned robotics cases?

R: As described in "Methods-Patient Population", 196 consecutive patients were underwent robot-assisted TT and concomitant CCND using a gasless trans-axillary approach for PTC between December 2008 and September 2017. After excluding two patients with unavailable data, 194 patients were enrolled. Patients who were excluded from the present study were 53rd and 190th patients. These two patients had insufficient data associated with surgical completeness, including postoperative anti-Tg antibody levels. This was why we excluded these two patients from the present study.

10.   In figure 2, the RA-susum value is decreasing after 119th case. Is it due to the surgeon get familiar to the robotic system and the risk value P_i is estimated lower?

R: As described in " Discussion ", the CUSUM curve for the incidence of temporary hypoparathyroidism peaked at the 119th case, which was close to the second peak on the operation time CUSUM curve at the 121st case. The surgical proficiency was gained by advanced anatomical understanding of feeding vessels, venous drainage from parathyroid glands, meticulous dissection, individual ligation, appropriate handling of energy devices, and ensuring a safe distance to avoid thermal injury to parathyroid glands. The surgical proficiency was obtained during the challenging phase and it probably contributed to the lower incidence of temporary hypoparathyroidism observed during the competent phase.

Reviewer 2 Report

1. There have been some innovations in the field of surgical robot. Da Vinci surgical systems has had several model changes; Standard(1999-), da Vinci S(2006-), da Vinci Si (2009-) and da Vinci Xi(2015-). There are no descriptions about the robot the authors had been using or model changes they might have experienced in the period of the study. This should be written in the part of Materials and Methods.

2. Why did the authors limit the pathological diagnosis of the case series to papillary thyroid carcinomas? There should have been some patients with follicular carcinomas or poorly differentiated carcinomas who received TT using robot. However this may not have to be written in the manuscript, I just want to know.

3. I would like to know the date about blood loss in robotic thyroid surgery, even if it may be very small amount. Please add “Blood loss” in Table 2.as another characteristic if the authors have the data.

4. I imagine there may be a few cases which were started using robot but were converted into open surgery because of some reasons such as massive bleeding in the initial learning period. This should be written in the manuscript, if any.

5. The authors described that an inclusion criterion was "minimal invasion to the anterior thyroid capsule and strap muscles" in cases before 2014 and that an exclusion criterion was "posterior capsular invasion particularly adjacent to the tracheoesophageal groove". Thanks to the progress of ultrasonography, it is not difficult to detect massive extrathyroid extension before surgery.

However, the primary tumors of more than 20% patients had extensive extrathyroid extension in looking at Table 1 and these cases might not fulfill the criteria of robotic surgery especially in cases before 2014.

6. As the authors mentioned, the 2015 ATA guideline recommends us more conservative surgical strategy and makes us consider lobectomy to the patients whose primary tumors are less than 4cm  in diameter. If the authors accepted this guideline during the study period, it looked strange that mean primary tumor size was 1.1cm  after the learning curve end-point (Table1)

7. The authors described that bilateral CCND was performed in the patients with bilateral cancer. However, the performance rates of bilateral CCND were less than half of the occurrence rates of bilateral cancers in Table 1. It does not make sense to me.

Author Response

Answers to reviewers’ comments

Thank you for giving us the opportunity to revise our manuscript. My colleagues and I feel that the quality of our article has greatly improved because of the reviewers’ insightful comments, and we have provided a ‘point-by-point’ reply to each comment below.

Reviewer 2

1. There have been some innovations in the field of surgical robot. Da Vinci surgical systems has had several model changes; Standard (1999-), da Vinci S (2006-), da Vinci Si (2009-) and da Vinci Xi (2015-). There are no descriptions about the robot the authors had been using or model changes they might have experienced in the period of the study. This should be written in the part of Materials and Methods.

R: Thank you for your insightful comments. All procedures were carried out by a single endocrine surgeon (JH Yoon) using the da Vinci Si or Xi intuitive surgical system (Sunnyvale, CA, USA). We have added what you commented in the ‘Materials and Methods’ section as the ‘Robotic procedure and instruments’.

2. Why did the authors limit the pathological diagnosis of the case series to papillary thyroid carcinomas? There should have been some patients with follicular carcinomas or poorly differentiated carcinomas who received TT using robot. However, this may not have to be written in the manuscript, I just want to know.

R: Thank you for your interest. We have only been applying the robotic surgery only to differentiated thyroid carcinoma so far. In the present study, excluding the patients with follicular thyroid carcinoma (FTC) was because most of FTCs were diagnosed after completing a diagnostic lobectomy for follicular neoplasm or atypia of undetermined significance on fine needle aspiration cytology. Even though, the diagnosis of FTCs was confirmed on final histopathology reports, the completion of thyroidectomy was performed only in patients having extensive vascular invasion. 

3. I would like to know the date about blood loss in robotic thyroid surgery, even if it may be very small amount. Please add “Blood loss” in Table 2.as another characteristic if the authors have the data.

R: Thank you for your insightful comments. We have added this data in the ‘Results’ section and the Table 2.

4. I imagine there may be a few cases which were started using robot but were converted into open surgery because of some reasons such as massive bleeding in the initial learning period. This should be written in the manuscript, if any.

R: Thank you for your comment. As we described in the ‘Results’ section, all robotic procedures were successfully completed without conversion to conventional open surgery.

5. The authors described that an inclusion criterion was "minimal invasion to the anterior thyroid capsule and strap muscles" in cases before 2014 and that an exclusion criterion was "posterior capsular invasion particularly adjacent to the tracheoesophageal groove". Thanks to the progress of ultrasonography, it is not difficult to detect massive extrathyroid extension before surgery.

However, the primary tumors of more than 20% patients had extensive extrathyroid extension in looking at Table 1 and these cases might not fulfill the criteria of robotic surgery especially in cases before 2014.

R: Thank you for your constructive comments. As you mentioned, the neck ultrasonography (US) now has a high resolution, but the findings on preoperative staging work-up might be quite different from intraoperative ones, especially in the cases in which the primary tumors are abutting in posterior portion of thyroid gland. Even though the primary tumors located posteriorly do not show definitive evidences of invasion on preoperative imaging studies, the invasion to recurrent laryngeal nerve, trachea, and esophagus might sometimes be found intraoperatively. However, these invasions that were incidentally found during operation are usually not neither aggressive nor wide, and can be shaved off without conversion to conventional open surgery. Metastatic central compartment lymph nodes (LNs) invading surrounding structures unidentified on preoperative staging work-up are also found during operation, because the accuracy of preoperative imaging studies for identifying metastatic central compartment LNs has been shown to be suboptimal.

6. As the authors mentioned, the 2015 ATA guideline recommends us more conservative surgical strategy and makes us consider lobectomy to the patients whose primary tumors are less than 100px in diameter. If the authors accepted this guideline during the study period, it looked strange that mean primary tumor size was 27.500000000000004px after the learning curve end-point (Table1)

R: Thank you for your comments. As we described, our institutional inclusion criteria have changed since 2014. However, most of patients were diagnosed as having small thyroid cancer through health screening tests, causing little changes in the mean primary tumor size.

7. The authors described that bilateral CCND was performed in the patients with bilateral cancer. However, the performance rates of bilateral CCND were less than half of the occurrence rates of bilateral cancers in Table 1. It does not make sense to me.

R: Thank you for your insightful comments. There were typographical errors. As we described in the ‘Materials and Methods’ section, at our institution, we always perform central compartment node dissection (CCND) in operations for patients with differentiated thyroid carcinoma even though there are no definite evidence of LN metastasis on preoperative staging work-up using neck US and computed tomography scans. This prophylactic CCND was usually performed unilaterally on the lesion side only unless there were enlarged LNs or LNs suspicious of metastases, even though the primary cancers are bilateral. Furthermore, occult carcinomas in contralateral lobe might be found on final histopathologic reports in cases in which the prophylactic unilateral CCND was performed for unilateral cancers diagnosed on preoperative evaluation. We have revised these in the ‘Materials and Methods’ section.

Round 2

Reviewer 2 Report

I checked the revised manuscript and point to point answers to my comments. Every comments I raised are well responded.